# The Drivers of the Sustainability of Spanish Wineries: Resources and Capabilities

**María Carmen García-Cortijo** [1,*] **, Juan R. Ferrer** [2] **, Juan Sebastián Castillo-Valero** [1] **and Vicente Pinilla** [3]

1 Instituto de Desarrollo Regional, Universidad de Castilla-La Mancha, Paseo de los Estudiantes s/n, 02071 Albacete, Spain; sebastian.castillo@uclm.es
2 ETSI Agronómica, Alimentaria and Biosistemas, Universidad Politécnica de Madrid, Av. Puerta de Hierro, nº 2, 4, 28040 Madrid, Spain; juanramon.ferrer@upm.es
3 Department of Applied Economics, and Instituto Agroalimentario de Aragon (IA2), Universidad de Zaragoza, Gran Vía, nº 2, 50005 Zaragoza, Spain; vpinilla@unizar.es
* Correspondence: mariacarmen.gcortijo@uclm.es

**Abstract:** This article aims to determine which of a firm's resources are drivers of its decisions on sustainability policies. For this purpose, it analyses four of the resources that the literature has linked with sustainability: (1) marketing resources, (2) technological resources, (3) innovation resources and (4) financial resources. The study focuses on Spain, which has the largest surface area under vine in the world. The database for the empirical analysis was drawn up from a survey among wineries carried out during 2020 and 2021. A total of 411 observations were valid. From the quantitative analysis, based on Box–Cox models, it can be concluded that adopting sustainability policies requires placing stress on innovation and on the capacity for communicating such innovations so that consumers perceive them as a change for the better; having greater technological or financial resources seems to be insufficient and of little importance. The results indicate that promoting funding and resource availability as basic tools should be reviewed in sustainability policies for wine firms.

**Keywords:** drivers; sustainability; resources; competitive advantage; organic wine; carbon footprint; corporate social responsibility

## 1. Introduction

Sustainability has become a key goal for many public, private, national and international institutions. It hovers over business activity, creating a new paradigm, that of searching for social and economic advances that will guarantee healthy and productive life for human beings without affecting the possibilities of future generations [1–6].

The wine sector, like any other, has to face the challenge of moving towards sustainability [1,7,8]. However, it has certain characteristics that make its approach different, such as: (1) vines are often grown on land where other crops would not flourish [9,10]; (2) it supports the population of rural areas and allows for the creation of wealth and of jobs in local economies [11]; (3) it is based on values relating to family and culture [12]; (4) it has a long tradition behind it and requires time and the transmission of values [9,13–15]. In parallel, activity in the wine sector generates unestimated costs or negative externalities, such as land use, water consumption, energy use, pesticides, wastewater, solid waste, as well as the carbon footprint stemming from its activity and especially from transport [7,12–14,16–19].

Sustainability involves three factors: economic viability, the environment and social acceptance [20,21]. However, these three elements are not always present in scientific studies or in the minds of consumers, entrepreneurs, or workers when they consider this concept [2,18]. Many studies or certificates that mention sustainability focus on the environmental aspects but neglect the other two [2,4,12,16,18]. The purpose of this article is to determine which of a firm's resources are the ones that drive its decisions on sustainability policies. We consider the wine sector in Spain, on which there have been few

studies in this connection, except for some partial studies [6,13]. In order to capture the commitment to sustainability of the Spanish wineries, we analyse organic wine, the carbon footprint and CSR. With respect to the possible resources that can act as drivers of these factors that lead to sustainability, we will study the marketing, technological, innovation and financial resources.

### 1.1. Literature Review and Hypotheses

Sustainability appeared for the first time in 1987 in the Brundtland Report [2,3] as a concept linked to sustainable development, that is, one that aims to meet the needs of the present generation without affecting the capabilities of future generations. For some authors, to link development and sustainability is an oxymoron, in that development destroys the very roots of sustainability. This means that the definition of development is plagued by ambiguity or distortion [4,22]. Sustainable development has focused on ecological aspects when talking about preserving the environment, generating confusion [2,3,7,12,13,23], especially considering that international institutions, such as the European Union (EU), the World Bank (WB), the Organisation for Economic Cooperation and Development (OECD) or the United Nations Development Programme (UNDP) treat sustainability from three viewpoints—economic, social and environmental [3].

In the wine sector, sustainability practices have been adopted increasingly since 2000 [8,24,25]. Sustainable winemaking was considered in 2004 by the International Organisation of Vine and Wine as a "global strategy on the scale of the grape production and processing systems, incorporating at the same time the economic sustainability of structures and territories, producing quality products, considering requirements of precision in sustainable viticulture, risks to the environment, products safety and consumer health and valuing of heritage, historical, cultural, ecological and aesthetic aspects". Subsequently, under point 1 of its general principles, the organisation states that sustainable development should be based on the ability to reconcile its three dimensions—economic, environmental and social [21].

There have been many articles about research on sustainability in the wine sector since 2002, but they do not consider what should, or should not, be included in sustainability. This gives rise to a multiplicity of concepts and classifications such as green business, green-green business, organic or sustainable production, sustainability, biodynamic production, ecopreneurship and environment [7,25].

In this study we will analyse three elements that the previous literature associates with sustainability: organic wine, the carbon footprint and corporate social responsibility [6,26,27].

We understand organic wine to be that which has been developed following the EU standards on organic agriculture. According to these standards, the production method should have the objective of obtaining wine using natural substances and processes. In this way, the environmental impacts are reduced due to the promotion of a responsible use of energy and natural resources, the maintenance of the biodiversity, the conservation of the regional ecological balances, the improvement of soil fertility and the maintenance of the quality of the water [26].

In this study, we understand the carbon footprint as a single-issue indicator commonly used to express the pressure of human activities on the environment. CF quantifies the impact of a given activity/process/product in terms of equivalent carbon dioxide ($CO_2eq$) emissions, considering the total amount of direct and indirect GHG emissions [27]. The importance of the carbon footprint resides in it being an indicator of environmental sustainability that quantifies the emissions of greenhouse gases generated during the lifecycle of a product.

Corporate social responsibility (CSR) is the responsibility that companies have for their impact on society (social, economic and environmental) and, therefore, seeks to minimise negative impacts and maximise positive ones. Sustainability refers to the company's ability to meet its needs without compromising future generations [28]. Corporate social responsi-

bility is often related to sustainability given that it is the way that enables companies to express their commitment to it [6].

When studying sustainability, it is important to find out what its drivers are, that is, the resources and strategies that allow for sustainable behaviour in firms. Knowledge of the drivers is a fundamental element in the development of sustainability in companies, given that through them and their implementation sustainable companies may be achieved. Drivers are usually classified as either external or internal [7,8]. External drivers are consumers, the market, policies and stakeholders who may set up initiatives in environmental matters such as reduced water consumption, impact on the community, the use of chemicals, waste management, land use, energy use and the greenhouse effect [12]. However, several studies have found that internal drivers are the most important for the adoption of sustainability practices [8,12,17]. Internal drivers include strategic decisions based on ethical reasons, operational efficiency, market positioning, personal values, professional preferences and satisfaction and product quality [7,12,17]. However, there have been very few studies linking a firm's drivers with its sustainability even though some authors have indicated that the lack of certain resources, such as funding, might hold back sustainability in wineries [8].

In this article, we will focus on resources and capabilities as drivers of sustainability. This approach to sustainability is novel and is based on the philosophy of the search for a comparative advantage. This same approach has been contemplated in other studies that have related the business model with sustainability [29,30]. One study examines 106 companies in Italy and the influence of the BM of the family or non-family business on sustainability [30]. Another analyses the relationship between sustainability and the BM, based on a qualitative analysis of 11 wineries in France and Italy and examines the business trend towards sustainability and the level of performance, resources, innovation and value created [29].

The theory of resources and capabilities indicates that the availability of strategic resources and capabilities is key to achieving a competitive advantage [31]. In this theory, resources are all the factors available to the firm, and capabilities are developed over time on the basis of complex interactions between them [32].

Of the various resources, those included in this study are the ones that the literature relates to sustainability: marketing resources, technological resources, innovation resources and financial resources.

The availability of marketing resources allows firms to present consumers with products that are environment-friendly, socially acceptable and potentially profitable for the firm. Consumers of wine care about sustainable products [12,13,18] but value them differently depending on the country and the market segment [23,33,34]. The availability of marketing capabilities allows a product's characteristics to be communicated and differentiated, reaching potential consumers [23,33,35–37].

Technology allows for organic production and lower emission of gases that are harmful for the atmosphere and is key for taking actions for sustainability [38,39]. The existence of technology makes it possible to set goals to reduce environmental impacts, to present products that are accepted by society and to promote better use of resources. Wineries that do not have technology cannot adopt this type of policy [40–42].

Innovation resources are based on the capability of a firm to improve its product, processes and business organisation [43]. To achieve sustainability and reduce impacts, firms must have new and innovative processes and varieties [38,40,44]. Innovation thus becomes a key element in a business model that aims to achieve sustainability in the wine sector [43].

For a firm to work towards sustainability, it also needs financial resources [8] so that it can carry out actions that include reducing pesticide use, land management, lower water and energy consumption, effluent and waste treatment, communication to society and searching for higher economic returns [9,44].

A firm's competitive advantage, resources, value created and innovation all affect its sustainability [8,17,44,45]. All the above leads us to pose two hypotheses, the first of which is divided into four sub-hypotheses, as follows:

**Hypothesis 1a (H1a).** *Wineries that have more resources will be more likely to adopt a sustainability policy.*

**Hypothesis 1b (H1b).** *Marketing resources.*

**Hypothesis 1c (H1c).** *Technological resources.*

**Hypothesis 1d (H1d).** *Innovation resources.*

**Hypothesis 1e (H1e).** *Financial resources.*

**Hypothesis 2 (H2).** *Wineries that have a competitive advantage will be more likely to adopt a sustainability policy.*

*1.2. Case Study. Spanish Wine Sector*

Spain has been selected as a case study. It has a larger surface area under vine than any other country in the world—961 mha in 2020. Wine production during 2020–2021 amounted to 40.3 mhl. Of this, 85% was sold as bulk wine and 15% was bottled, using the stocks at 1 June 2021 as an indicator [46]. It is the world's third largest producer with 40.3 mhl, after Italy (49.1 mhl) and France (46.6 mhl). It is also the country that saw the greatest increase in production volume in 2020 over 2019 with 7.0 mhl (+21%), as opposed to 1.5 mhl (+3%) in Italy and 4.4 mhl (+11%) in France. Moreover, Spain's production volume in 2020 grew by 8% over its last 5-year average, while the Italian volume remained similar and for France the increase was +6% [15].

Regarding consumption, there was a marked downward trend after the mid-1960s, when the peak of 70 L per capita was reached. Today the figure is about 15 L [47]. In recent years, consumption in Spain has dropped in absolute terms from 14 mhl in 2000 to 9.6 mhl in 2020 [15]. This drop in consumption has forced wineries to sell a large proportion of their production outside Spain [48].

Spain's organic wine is attracting increasing interest in the context of more sustainable agriculture. The area growing organic wine in Spain grew by 8% in 2020 to 131,183 hectares, that is 14% of total vineyards and 26.88% of global organic vineyards. Spain is thus in the lead for organic vineyards, ahead of Italy, France and China. This type of production has grown constantly in recent years: from 2009 to 2020, the area almost tripled—from 53,958 ha to 131,183 ha. Additionally, the number of wineries producing organic wine rose from 408 to 1214, accounting for 14% of all wineries [49].

However, work is still needed on certification, which is one of the elements that have proved to be relevant for consumers so that they can recognise an organic or sustainable wine. The Spanish Wine Federation (Federación Española del Vino, FEV) is promoting an accreditation of sustainability named "Wineries for Climate Protection" which, in May 2021, was held by 32 wineries [50]. There is also an association of small wineries called Spanish Organic Wine which aims to help its members sell their products abroad in view of the difficulties of the domestic market. In May 2021, there were 39 members [51]. It therefore seems that, even though much has been done on the path towards sustainability, certification and accreditation need to progress further in order to position wines and wineries as sustainable, allowing them to enjoy the advantages of recognition by consumers [23].

The business structure of the Spanish winemaking sector is highly atomised. On 1 January 2020, there were 4133 registered wineries (CNAE Code (National Classification of Economic Activities) 1102: Winemaking). The number of wineries registered has decreased slightly since 2018 when there was a maximum of 4052 [52].

The Spanish companies engaged in winemaking are eminently family-run businesses and of a small size. In 2018, 27.3% of them had no employees (latest available data); and 84.7% of total wineries had less than 10 employees. The most frequent legal personality in wine sector companies is that of the limited liability company, with almost half having this

status. This is followed by other forms of legal personality, such as the natural person, the limited company and cooperative business, all accounting for between 15% and 19% of the total [52].

Total labour costs in Spanish wineries grew from EUR 762.5 M to EUR 979 M between 2015 and 2019, which implies a growth of more than 200 people and over 28% in five years. Turnover has increased from EUR 5346 M in 2009 to a little over EUR 8000 M in 2019. It represents 6% of the turnover of the food and drinks industry, contributes 8% of the gross added value of this industry and employs 7% of its workers. The leading Spanish winery in terms of sales in 2019 was Freixenet, with EUR 170.08 million in turnover, followed by Miguel Torres (EUR 169.87 M) and Félix Solís (EUR 148.31 M) [52–55].

Seventy-one percent of the wineries are exporters. In the complicated year of 2020, 3536 companies exported wine worth a value of EUR 2687.4 M. With respect to 2019, the number of exporting companies reduced by 4% (−157). In 2020, turnover also fell by 3% to EUR 2687.4 M (−EUR 93.1 M). With respect to the segmentation in accordance with the amount billed for wine exports, many companies exported a small quantity of the total, while a few exported the majority with 71.3% of total exports. This high percentage of exports was carried out by just 3.1% of the exporting wine companies (111 companies) with an average of over EUR 5 M in sales per company. Europe is the continent where the most Spanish wine is exported, representing 67.6% of total turnover (EUR 1815.7 M), with 2131 exporting companies. Within Europe, the Euro zone accounts for 40.4% of total sales, followed by the rest of the EU (7% of the total) and the rest of Europe (13% of the total) [56].

After providing an overall view of the sector in Spain, we will now address the principal objective of our study, which is to determine which factors influence the preferences of the wineries in Spain for certain sustainability measures or others, based on the survey that we have conducted among the Spanish wineries.

## 2. Materials and Methods

### 2.1. Sample and Variables

The data used in this article have been obtained from a survey conducted among companies operating in Spain and whose economic activity is winemaking. The data gathering process was carried out during 2020 and the first three months of 2021. After sending out the questionnaire via e-mail to the managers of the firms [57,58], the authors waited for a month to receive replies. In cases where there was none, they called recipients to remind them. The final sample comprised 411 valid answers out of 2977, which amounts to a rate of response of 14% for the industrial sector [59]. We therefore consider it suitable for our study. This figure does not entail problems of significance for the statistical results because the sample error is 0.045.

The questionnaire was conducted after a literature review and used scales that had been validated in previous studies. The items of the survey focused on the resources and capabilities, the competitive environment, the business strategy, business performance and sustainability preferences. Next, we will present the items that will be the dependent variables in our models.

### 2.1.1. Dependent Variable

The purpose of our study is to analyse whether the resources held by a firm and its possible competitive advantage are drivers of sustainability. We used three dependent variables, all of which have been related to sustainability: (1) production of organic wines in the winery [13,34]; (2) concern for the firm's carbon footprint [1,18]; and (3) corporate social responsibility [6]. The methodology for formalising each dependent variable is given in Table 1.

**Table 1.** Description of the variables used.

| Variables | Typology | Description |
|---|---|---|
| Organic wine ($Y_{VE}$) | Continuous | The wineries valued their interest in organic wine as an environmental measure on a scale from 1 to 5, with 1 for very low interest and 5 for very high interest |
| Carbon footprint ($Y_{HC}$) | Continuous | The wineries valued their interest in calculation of their carbon footprint on a scale from 1 to 5, with 1 for very low interest and 5 for very high interest |
| Corporate responsibility ($Y_{RSC}$) | Continuous | The wineries estimated their interest in adopting this measure on a scale from 1 to 5, with 1 for very low interest and 5 for very high interest |
| Legal status (SJ) | Discrete | The variable takes 1 if the firm is a cooperative, and 0 otherwise |
| Size (SI) | Discrete | Number of employees<br>Takes 1 if the firm is a micro or small enterprise (less than 50 workers)<br>Takes 2 if the firm is medium-sized (between 50 and 249 workers)<br>Takes 3 if the firm is large (over 250 workers) |
| Exports (X) | Discrete | Takes 1 if the winery exports, 0 otherwise |
| Marketing resources (EM) | Discrete | Takes 1 if the firm has a much worse position than the competition<br>Takes 2 if the firm has a worse position than the competition<br>Takes 3 if the firm has a similar position to the competition<br>Takes 4 if the firm has a better position than the competition<br>Takes 5 if the firm has a much better position than the competition |
| Technological resources (RT) | Discrete | Takes 1 if the firm has a much worse position than the competition<br>Takes 2 if the firm has a worse position than the competition<br>Takes 3 if the firm has a similar position to the competition<br>Takes 4 if the firm has a better position than the competition<br>Takes 5 if the firm has a much better position than the competition |
| Innovation resources (IN) | Discrete | Takes 1 if the firm has a much worse position than the competition<br>Takes 2 if the firm has a worse position than the competition<br>Takes 3 if the firm has a similar position to the competition<br>Takes 4 if the firm has a better position than the competition<br>Takes 5 if the firm has a much better position than the competition |
| Financial resources (PF) | Discrete | Takes 1 if the firm has a much worse position than the competition<br>Takes 2 if the firm has a worse position than the competition<br>Takes 3 if the firm has a similar position to the competition<br>Takes 4 if the firm has a better position than the competition<br>Takes 5 if the firm has a much better position than the competition |
| Competitive advantage (ROA) | Discrete | Takes 1 if ROA is below 5%<br>Takes 2 if ROA is 5%–15%<br>Takes 3 if ROA is 15%–25%<br>Takes 4 if ROA is 25%–35%<br>Takes 5 if ROA is 35%–45%<br>Takes 6 if ROA is above 45% |

### 2.1.2. Independent Variables

Following prior studies on resource availability in firms, we used four variables that define the availability of marketing, technology, innovation and financial resources. The firm managers were asked to define their position for resources in relation to the competition on a scale from 1 to 5 [57,58], as shown in Table 1.

Competitive advantage is measured by business performance (ROA) [6], with the best-performing firms being those that have a competitive advantage [10,60], as shown in Table 1.

### 2.1.3. Control Variables

This study includes several control variables (see Table 1) which help explain the effects of the independent variables on the dependent ones [10,57,58].

The variables chosen were:

(1) Legal status, with firms taking 1 if they are cooperatives and 0 otherwise. Various authors relate cooperatives with sustainability [9,61,62], and this variable is expected to have a positive effect.

(2) Firm size, measured by the number of employees. Firm size is expected to have a positive effect on sustainability because a larger size makes it easier to devote resources and employees to such policies [29].

(3) Exports, with firms taking 1 if they export and 0 otherwise. International markets increasingly place value on sustainable products presented by environment-friendly firms [23,31], so this variable is expected to have a positive effect.

### 2.2. Functional Form

For this study, we used a Box–Cox regression model because the dependent variables ($Y_i$) do not follow normal distribution with $p < 0.05$ for the Shapiro–Wilk W test: W = 0.98542 ($p = 0.00055$) for $Y_{VE,i}$, W = 0.97812 ($p = 0.00001$) for $Y_{HC,i}$, W = 0.99317 ($p = 0.07326$) for $Y_{RSC,i}$. Since the independent variables are discrete, we applied the lhsonly left-hand-side Box–Cox model.

We drew up a model for each measure of sustainability $Y_{VE,i}$, $Y_{HC,i}$, $Y_{RSC,i}$, analytically expressed as follows:

$$\begin{aligned}
Y_{VE}{}^\theta{}_i &= \beta_0 + \beta_1 SJi + \beta_2 SI_i + \beta_3 X_i + \beta_4 RM_i + \beta_5 RT_i + \beta_6 RI_i + \beta_7 RFi + \beta_8\, ROA_i + u_i \\
Y_{HC}{}^\theta{}_i &= \beta_0 + \beta_1 SJi + \beta_2 SI_i + \beta_3 X_i + \beta_4 RM_i + \beta_5 RT_i + \beta_6 RI_i + \beta_7 RFi + \beta_8\, ROA_i + u_i \\
Y_{RSC}{}^\theta{}_i &= \beta_0 + \beta_1 SJi + \beta_2 SI_i + \beta_3 X_i + \beta_4 RM_i + \beta_5 RT_i + \beta_6 RI_i + \beta_7 RFi + \beta_8\, ROA_i + u_i
\end{aligned} \tag{1}$$

where $u_i \sim N(0, \sigma 2)$.

The dependent variables are subject to theta $\theta$ transformation: $Y_{VE}{}^\theta{}_i$, $Y_{HC}{}^\theta{}_i$, $Y_{RSC}{}^\theta{}_i$. The independent variables are: legal status (SJ), size (SI), exports (X), marketing resources (RM), technological resources (RT), innovation resources (RI), financial resources (RF) and performance (ROA). ui is the random disturbance.

## 3. Results

First, we identified the value of $\theta$ using the lhsonly left-hand-side Box–Cox model, selecting the power $\theta$ with a *p*-value above 0.05 for the LR test associated with $\theta$ with values (−1, 0, 1) (Table 2), and below 0.05 for specific $\theta$ values (Table 3).

**Table 2.** LR statistic for powers with theta values (−1, 0, 1).

| | LR Statistic Test h0 | Restricted Log Likelihood | LR Statistic Chi2 | *p*-Value Prob > chi2 |
|---|---|---|---|---|
| Organic wine. lhsonly left-hand-side Box–Cox model | theta = −1 | −687.29937 | 355.37 | 0.000 |
| | theta = 0 | −560.57392 | 101.92 | 0.000 |
| | theta = 1 | −511.15306 | 3.08 | 0.079 |
| Carbon footprint. lhsonly left-hand-side Box–Cox model | theta = −1 | −648.21062 | 430.72 | 0.000 |
| | theta = 0 | −504.14001 | 142.57 | 0.000 |
| | theta = 1 | −441.85035 | 17.99 | 0.000 |
| CSR. lhsonly left-hand-side Box–Cox model | theta = −1 | −628.76422 | 334.28 | 0.000 |
| | theta = 0 | −506.28738 | 89.33 | 0.000 |
| | theta = 1 | −462.9078 | 2.57 | 0.109 |

**Table 3.** Theta powers estimated by the Box–Cox procedure.

| | Power | Std. Coeff. | Err. | z | *p* > z |
|---|---|---|---|---|---|
| Organic wine. lhsonly left-hand-side Box–Cox model | theta | 1.226424 | 0.1312753 | 9.34 | 0.000 |
| Carbon footprint. lhsonly left-hand-side Box–Cox model | theta | 1.648547 | 0.1632566 | 10.10 | 0.000 |
| CSR. lhsonly left-hand-side Box–Cox model | theta | 1.227442 | 0.1449591 | 8.47 | 0.000 |

For organic wine, the lhsonly left-hand-side Box–Cox model determines two possible values for theta: $\theta = 1$ and $\theta = 1.226424$; also, for corporate social responsibility: $\theta = 1$ and $\theta = 1.227442$. We therefore resorted to the lower mean squared error to select theta, which for organic wine was $\theta = 1.226424$ (squared error of 79.35 as opposed to 498.39 for $\theta = 1$)

and for corporate social responsibility θ = 1.227442 (squared error of 62.20 as opposed to 359.67 for θ = 1). For carbon footprint, the value was θ = 1.648547.

The models estimated were correctly specified according to the RESET test. F-Snedecor, with a *p*-value below 0.05, shows the global capacity of all the model's explanatory variables. VIF is lower than 10 so there is no multicollinearity in the models. However, the White test, with a *p*-value above 0.05 points to the absence of heteroskedasticity in the models, so random disturbance maintains the same dispersion for all the observations (Table 4).

**Table 4.** Estimation results.

| | Organic Wine | Carbon Footprint | CSR |
|---|---|---|---|
| Legal status (SJ) | 0.0976672 * (1.784) | 0.247969 (0.6527) | −0.00436625 (−0.08182) |
| Size (SI) | −0.0736889 (−1.355) | 0.131485 (0.3455) | −0.00777950 (−0.1484) |
| Exports (X) | 0.295236 *** (3.201) | 0.220904 (0.3776) | 0.0834366 (0.9987) |
| Marketing resources (RM) | 0.0836874 *** (2.615) | 0.662834 *** (2.860) | 0.0841469 ** (2.583) |
| Technological resources (RT) | −0.0229471 (−0.6048) | −0.178991 (−0.6152) | −0.0357241 (−0.8849) |
| Innovation resources (RI) | 0.0141772 (0.3713) | 0.548647 ** (2.095) | 0.0856231 ** (2.401) |
| Financial resources (RF) | −0.0510630 (−1.703) | −0.0443941 (−0.2041) | 0.0450384 * (1.695) |
| Competitive advantages (ROA) | 0.0462017* (1.699) | 0.341189 * (1.776) | 0.0444551 * (1.632) |
| Const. | 1.60319 *** (12.45) | 3.34305 *** (3.601) | 1.46148 *** (11.01) |
| **RESET specification test** | F = 0.829297, with $p$ = p (F (2297) > 0.829297) = 0.437 | F = 0.076590, with $p$ = p (F (2303) > 0.0765899) = 0.926 | F = 1.650336, with $p$ = p (F (2299) > 1.65034) = 0.194 |
| **F-Snedecor** | F(8, 299) = 4.958804 p (of F) $8.85 \times 10^{-6}$ | F(8, 305) = 3.975080 p (of F) 0.000169 | F(8, 301) = 5.292020 p (of F) $3.20 \times 10^{-6}$ |
| **Sum of residuals squared** | 79.35785 | 30.05 | 62.20745 |
| **Variance inflation factors (VIF)** | SJ: 1.101, SI: 1.170, X: 1.098, EM: 1.298, RT:1.807, IN: 1.746, RF: 1.197, ROA:1.071 | SJ: 1.101, SI: 1.170, X: 1.098, EM: 1.298, RT:1.807, IN: 1.746, RF: 1.197, ROA:1.071 | SJ: 1.101, SI: 1.170, X: 1.098, EM: 1.298, RT:1.807, IN: 1.746, RF: 1.197, ROA:1.071 |
| **White heteroskedasticity test** | LM = 46.6712 with $p$ = p (Chi-squared (42) > 46.6712) = 0.286347 | LM = 33.8447 with $p$ = p (Chi-squared (42) > 33.8447) = 0.810814 | LM = 34.5667 with $p$ = p (Chi-squared (42) > 34.5667) = 0.785404 |

In brackets, the t-statistics of the coefficient estimates. * Denotes significance at the 10-percent level, ** denotes significance at the 5-percent level, *** denotes significance at the 1-percent level.

The results show that available resources are important, but their weight varies with the dependent variable analysed (organic wine, carbon footprint and CSR).

For organic wine, the most important drivers are exports, marketing resources, legal status and competitive advantage; for carbon footprint, marketing resources, innovation resources and competitive advantage; and for corporative social responsibility, marketing resources, innovation resources, financial resources and competitive advantage. Neither size nor technological resources are significant in any case.

The study rejects hypothesis H1.2 because no relation was found between technological resources and sustainability variables. Hypothesis H1.3 is partially accepted because a relationship was found between innovation resources and two of the three sustainability variables studied (carbon footprint and CSR). Hypothesis H1.1 is accepted because in all three cases the relation with marketing resources was significant. Hypothesis H1.4 is partially rejected because the relation was only significant in the case of CSR. Finally, hypothesis H2 is accepted; in all cases a relation was found between competitive advantage and the adoption of sustainability policies.

## 4. Discussion

The purpose of this study was to assess the effect of resources and of competitive advantage as drivers of sustainability policies in wine firms in Spain. We analysed four of the resources that the literature has related to sustainability: marketing, technological, innovation and financial resources. Sustainability policies were assessed on the basis of the winery's intention to continue or adopt actions in three areas: production of organic wine, reduction of the carbon footprint and corporate social responsibility. We also used a number of control variables such as firm size, legal form (cooperative or not cooperative) and whether the firm exports or not.

The results show that only some resources are drivers of sustainability. Marketing resources played a relevant role in all three lines studied. The importance of marketing has already been pointed out by several authors and is based on the need for consumers to know the efforts made by the firm to present a differentiated product that is both sustainable and ecofriendly [2,12,23].

Another of the resources that facilitates the adoption of sustainability plans and policies is innovation, which was seen to be positive for reducing the carbon footprint and for CSR, but not for organic wine. It seems that having differential resources in innovation is not necessary to elaborate organic wine; in fact, in many cases in the elaboration of organic wine, the wineries go back to traditional practices. The importance of innovation has already been shown in prior studies that are closely related to the definition of a business model aiming to move the firm towards sustainability [8,30]. However, in this study, no clear relation was found between technological and financial resources with regard to plans for sustainability, in contradiction with prior studies that pointed to the importance of technology for sustainability [40–42]. The disposition of technological resources in the wine sector refers to the existence of equipment and facilities for carrying out the activity, the existence of production departments and economies of scale [63]. However, it does not mean that this equipment is modern, innovative and ecofriendly, and some authors differentiate between old and new technology, identifying new technology as the driver of sustainability [64]. It seems that it is not sufficient to simply have technology, but this technology must be innovative in order to be a driver of sustainability [65].

It seems that the adoption of sustainability policies depends more on intent and on innovation than on the availability of a specific technology. Something similar occurs with financial resources which, in this study, are not seen to be relevant for sustainability policies, except for CSR. This is in contradiction with prior studies which accepted that sustainability increases a firm's costs and requires greater funding [8,9,29]. Access to financial resources assists the company in making investments and improvements in the different areas of its activity, but this does not mean that these investments are directed towards sustainability. The availability of financial resources does not imply a greater inclination to implement sustainable policies. This has already been mentioned in a study of small-sized firms in Thailand [64]. In the case of CSR, the situation is somewhat different, as many companies see CSR as an element of distinction, legitimacy and reputation and it is found to be related to larger-sized companies with a higher amount of financial resources at their disposal [65].

It can be concluded that sustainability depends more on the intent of firms, providing they have innovations allowing them to adopt new sustainability practices, mainly in carbon footprint and CSR, and on good communication or marketing resources that will help them reach consumers. This element is important because the literature confirms that consumers are confused about differences between terms such as organic, biodynamic or sustainable wine or corporate social responsibility [12,13,18].

Finally, this study shows that having a competitive advantage is related to the adoption of sustainability policies. Competitive advantage is measured by having an ROA that is above that of the competition, thus guaranteeing economic viability [10,59]. This is of interest because other studies have not corroborated it [6], but it must be remembered that the actual definition of the concept of sustainability covers three factors: economic viability, environment and social acceptance [20]. It corroborates the fact that a better economic

situation allows firms to look towards a future that will undoubtedly be linked to the paradigm of sustainability [1–4,6,66].

## 5. Conclusions

This study adopted a novel approach to the study of sustainability in the wine sector. It was based on the analysis of the resources and capabilities of firms as drivers for sustainability policies. Although drivers have already been analysed by prior studies, resources had only been studied tangentially as part of a firm's business model. From the analysis carried out, it can be concluded that innovation and marketing are more important than technology and financial resources. It can therefore be deduced that in order to adopt sustainability policies, stress should be placed on innovation, mainly in carbon footprint and CSR, and on the capacity for communicating such innovations so that consumers perceive them as a change. However, it does not seem to be sufficient or important to have greater technological or financial resources, though it is necessary that the company must have a willingness to direct these resources towards sustainability. A lack of funding does not seem to be a basic determinant for holding back sustainability policies in wineries (although it is for development and implementation). This allows us to conclude that public regulation of incentives for R&D+I in this sector should be adjusted.

This study is an initial examination in our database of the dynamic of investments in sustainability. Although the size of the sample is representative of the sector and includes the whole of the wine sector in Spain, the study could be enhanced by including more areas relating to sustainability, environmentally differentiated wine, the carbon footprint and corporate social responsibility. The scope of analysis could be expanded, and this will be the next area of research to be undertaken.

**Author Contributions:** Conceptualisation: M.C.G.-C., J.R.F., J.S.C.-V. and V.P. Methodology, data collection and data analysis: M.C.G.-C. and J.R.F. Data presentation, writing, reviewing and editing: M.C.G.-C., J.R.F., J.S.C.-V. and V.P. All authors have read and agreed to the published version of the manuscript.

**Funding:** This study has been partially funded by the ERDF-Interreg SUDOE Project SOE3/P2/F0917, VINCI (Wine, Innovation and International Competitiveness), the Ministry of Science and Innovation of the Spanish Government (project PGC2018-095529-B-I00) and by the Department of Science, Innovation and Universities of the Government of Aragon (Research Group S55_20R).

**Data Availability Statement:** Data sets analysed during the study are available from the authors on reasonable request.

**Conflicts of Interest:** The authors declare no conflict of interest.

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
