# Peer review of "The Drivers of the Sustainability of Spanish Wineries: Resources and Capabilities"

_sustainability, doi:10.3390/su131810171_

Round 1

Reviewer 1 Report

The publication does not have enough study of the data of the Spanish wineries so I do not see that it is methodologically well structured for this magazine.

Author Response

Referee # 1 (The corrections that respond to comments by this Reviewer are shown in blue in the main document)

Point 1: The publication does not have enough study of the data of the Spanish wineries so I do not see that it is methodologically well structured for this magazine.

Following the suggestions of the referee, we have expanded on the business structure of the Spanish wine sector: Number of wineries, evolution, sales, types of wineries, exports in the section on the case study.

Reviewer 2 Report

The article “Resources and sustainability in the Spanish wine sector” adopts a novel approach to the study of sustainability in the wine sector.

I suggest to add more information about organic wine, carbon footprint, corporate social responsibility, the three dependent variables related to sustainability and to explain why they are important factors for sustainability.

For the state of the art  I suggest: Science of the Total Environment, 759, 2021, 143462

https://doi.org/10.1016/j.scitotenv.2020.143462

In this study no clear relation was found between technological and financial resources with regard to plans for sustainability, in contradiction with other studies that pointed to the importance of technology. Also, for organic wine and carbon footprint size the technological resources are not significant. From the analysis carried out, it has been concluded that innovation and marketing are more important than technology and financial resources. How the authors can explain this? More correlations with literature should be done at discussion.

Author Response

Referee # 2 (The corrections that respond to comments by this Reviewer are shown in green in the main document)

Point 1: The article “Resources and sustainability in the Spanish wine sector” adopts a novel approach to the study of sustainability in the wine sector. I suggest to add more information about organic wine, carbon footprint, corporate social responsibility, the three dependent variables related to sustainability and to explain why they are important factors for sustainability.

For the state of the art I suggest: Science of the Total Environment, 759, 2021, 143462. https://doi.org/10.1016/j.scitotenv.2020.143462

We have enlarged the information about organic wine, carbon footprint and corporate social responsibility. For the state of the art, we have taken into account and cited the reference suggested by the referee.

Point 2: In this study no clear relation was found between technological and financial resources with regard to plans for sustainability, in contradiction with other studies that pointed to the importance of technology. Also, for organic wine and carbon footprint size the technological resources are not significant. From the analysis carried out, it has been concluded that innovation and marketing are more important than technology and financial resources. How the authors can explain this? More correlations with literature should be done at discussion.

Following the indications of the referee, we have expanded and improved the explanation of the results of the study.

Reviewer 3 Report

I would like to congratulate the authors for their interest in researching in this field, however, the work presented presents some deficiencies.

1. The title is not appropriate for the article. The title of the article provides little information about its content. I suggest that the authors modify the title to make it more illustrative.

The title should have the following characteristics:

-Describe the content of the article in a specific, clear, accurate, brief, and concise manner.

-Enable the reader to identify the topic easily.

-Allow a precise indexing of the material.

I suggest that the authors modify the title to make it more illustrative.

2. In the introductory chapter (chapter 1.2), the authors focus on the Spanish vineyard area. Although this data is representative, it should be contrasted with wine production and sales of bottled and bulk wine. The inclusion of these data would considerably enrich the introductory chapter, allowing the reader to have a global and complete vision of this sector in Spain.

3. The origin of the data obtained is not clearly defined in the document. The authors cite that 411 observations were made to obtain the baseline data. They also cite that the data were obtained from surveys held during 2020 and 2021. However, there is no description of the survey conducted and it is difficult to understand the data acquisition process.

4.  In addition to the previous point, the article implies that part of the data is obtained from public databases (economic data). I would ask the authors to clarify this point.

5. Line 287 “The text continues here.” ¿?¿? I do not understand the meaning of this phrase.

6. There are some methodological similarities with existing articles such as "Factors That Determine Innovation in Agrifood Firms," Agronomy, Agronomy, Inc. Innovation in Agrifood Firms", Agronomy, 2021.For this and future articles, I encourage the authors to explore the use of known methodologies for data quantification (LCA, S-LCA,...).

I hope that these changes will help to improve your article and make it a document of great scientific interest.

Author Response

Referee # 3 (The corrections that respond to comments by this Reviewer are shown in red in the main document)

Point 1:  The title is not appropriate for the article. The title of the article provides little information about its content. I suggest that the authors modify the title to make it more illustrative.

The title should have the following characteristics:

-Describe the content of the article in a specific, clear, accurate, brief, and concise manner.

-Enable the reader to identify the topic easily.

-Allow a precise indexing of the material.

I suggest that the authors modify the title to make it more illustrative.

Following the suggestions of the referee, the new title is:

THE DRIVERS OF THE SUSTAINABILITY OF SPANISH WINERIES: RESOURCES AND CAPACITIES

Point 2: In the introductory chapter (chapter 1.2), the authors focus on the Spanish vineyard area. Although this data is representative, it should be contrasted with wine production and sales of bottled and bulk wine. The inclusion of these data would considerably enrich the introductory chapter, allowing the reader to have a global and complete vision of this sector in Spain.

This information has been completed with data from the latest INFOVI report dated July 2021 (https://www.mapa.gob.es/es/agricultura/temas/producciones-agricolas/vitivinicultura/default.aspx). Following the suggestion of the referee, wine production has been broken down into bulk and bottled wine, but using the variable of stocks as an indicator, as production as such does not appear in disaggregated form, except for the type of wine according to its colour.  

Point 3: The origin of the data obtained is not clearly defined in the document. The authors cite that 411 observations were made to obtain the baseline data. They also cite that the data were obtained from surveys held during 2020 and 2021. However, there is no description of the survey conducted and it is difficult to understand the data acquisition process.

We thank the referee for this comment. This point is clarified by explaining that an online survey was carried out among all of the Spanish wineries and the number of responses obtained.

Point 4:  In addition to the previous point, the article implies that part of the data is obtained from public databases (economic data). I would ask the authors to clarify this point.

The data used are drawn from a survey. This aspect has been clarified.

Point 5: Line 287 “The text continues here.” ¿?¿? I do not understand the meaning of this phrase.

The phrase “The text continues here.” Has been eliminated.

Point 6: There are some methodological similarities with existing articles such as "Factors That Determine Innovation in Agrifood Firms," Agronomy, Agronomy, Inc. Innovation in Agrifood Firms", Agronomy, 2021.For this and future articles, I encourage the authors to explore the use of known methodologies for data quantification (LCA, S-LCA,...).

The referee is right in that another methodology could have been used such as those indicated. However, the methodology used was chosen because it adapted to our research very well for two reasons: 1) It adapted to the characteristics of the data obtained from the surveys which were cross section; and 2) It fit well with our objective: to identify which factors are the most influential for the three measures of sustainability adopted by the wineries. We thank the referee for the suggestions made (LCA, S-LCA...) and will take them into account for future studies.

Round 2

Reviewer 1 Report

Among the modifications that the author has made, we see that it is sufficient for publication; although I could improve the conclusions.

Author Response

The corrections that respond to comments by this Reviewer are shown in yellow in the main document.